# Predictive models for short-term load forecasting in the UK's electrical grid

**Yusuf A. Sha'aban** [ID] *

Department of Electrical Engineering, University of Hafr Al Batin, Hafr Al Batin, Kingdom of Saudi Arabia

* shaaban@uhb.edu.sa

## Abstract

There are global efforts to deploy Electric Vehicles (EVs) because of the role they promise to play in energy transition. These efforts underscore the e-mobility paradigm, representing an interplay between renewable energy resources, smart technologies, and networked transportation. However, there are concerns that these initiatives could burden the electricity grid due to increased demand. Hence, the need for accurate short-term load forecasting is pivotal for the efficient planning, operation, and control of the grid and associated power systems. This study presents robust models for forecasting half-hourly and hourly loads in the UK's power system. The work leverages machine learning techniques such as Support Vector Regression (SVR), Artificial Neural Networks (ANN), and Gaussian Process Regression (GPR) to develop robust prediction models using the net imports dataset from 2010 to 2020. The models were evaluated based on metrics like Root Mean Square Error (RMSE), Mean Absolute Prediction Error (MAPE), Mean Absolute Deviation (MAD), and the Correlation of Determination ($R^2$). For half-hourly forecasts, SVR performed best with an R-value of 99.85%, followed closely by GPR and ANN. But, for hourly forecasts, ANN led with an R-value of 99.71%. The findings affirm the reliability and precision of machine learning methods in short-term load forecasting, particularly highlighting the superior accuracy of the SVR model for half-hourly forecasts and the ANN model for hourly forecasts.

## Introduction

Energy is an important economic driver in modern societies and also serves as a highly essential commodity for domestic and industrial applications [1]. With global population growth and the uptrend in the establishment of production and manufacturing industries in many countries, the demand for more electricity is ever-increasing while actual generation lags due to resource constraints [2]. Part of the problem is solved by supplementing the power system with renewable sources such as solar and wind power and other forms of clean energy such as nuclear. Adding renewable energy to existing traditional sources means of power generation has the potential to increase energy output and efficiency [1, 3]. However, it also increases the complexity of balancing the load demand and actual generation [3]. The imbalance between electricity generation and load demands can cause serious grid disruptions [4]. Therefore, optimal balancing between electricity generated and load demand must be maintained to avoid disruptions. Devising operational strategies for power generation, transmission, and

**Data Availability Statement:** The data used for this work is publicly availabale for the sources given below: 1. Wilson IG, Sharma S, Day J, Godfrey N. Calculating Great Britain's half-hourly electrical demand from publicly available data. Energy Strategy Reviews. 2021;38:100743. doi:10.1016/j.

esr.2021.100743. 2. Wilson G. Electrical half hourly raw and cleaned datasets for Great Britain from 2008-11-05 (8.0.2) [Data set]; 2023. Available from: https://zenodo.org/records/7995513.

**Funding:** The author(s) received no specific funding for this work.

**Competing interests:** The authors have declared that no competing interests exist.

distribution networks also relies on accurate prediction of electric load [5, 6]. Power plant scheduling and unit commitment cannot be efficiently and reliably managed without accurate load forecasting [7]. Additionally, accurate load forecasting can be used for renewable energy sizing [8] and layout [9, 10].

Additionally, the move towards e-mobility further complicates the dynamics between the generation and demand for electricity. As Electric Vehicles (EVS) are integrated into the transport sector, they place a huge burden on the existing power grid and infrastructure. Hence, there have been research efforts to mitigate these challenges and allow for proper planning to accommodate the complexities introduced by the adoption of renewable energy resources (RES) and EVs in the energy mix and load demand interplay. RES are known to be intermittent and the charging needs for EVs depend on user behavior and other factors, hence further making the optimal solution to this problem complex. Moreover, when technologies such as vehicle-to-grid (V2G) operations are considered, the EVs can also be consolidated into a dispatchable energy source. Hence, there is a need for both short and long-term load forecasts to allow for stability enhancement, proper planning, maintenance, and optimal deployment of the available RESs [11–13]. In [14] for example, a method of coordinating EVs for bidirectional energy was presented. The scheme showed that upto 63% net savings in the cost of electricity for EV users. These gains will be more if real-time load forecasts are incorporated into the scheme.

The significance of precise load forecasting extends to operational cost estimation and profit maximization. A marginal 1% reduction in Mean Absolute Percentage Error (MAPE) can result in a 0.1-0.3% decrease in generation costs [5, 15, 16]. Previous studies indicate that a 1% increase in prediction error could escalate operational costs by approximately 10 million pounds annually in the British thermal power system [5, 15, 16]. Thus, the criticality of accurate forecasting cannot be over emphasized.

Load forecasting is broadly divided into four key subcategories based on the prediction time scale: Very Short-Term Load Forecasting (VSTLF), Short-Term Load Forecasting (STLF), Medium-Term Load Forecasting (MTLF), and Long-Term Load Forecasting (LTLF) [17, 18]. LTFL is applicable for the planning of power grids and generators and the forecast period spans a few months to several years. On the other hand, MTLF is useful for maintenance scheduling and the forecast period is between a few days to a few months. The STLF relates to generation and storage scheduling and the forecasting period ranges from a few hours to a few days. Finally, the VSTLF is needed for prevention and emergency control and its forecasting period is between a few seconds to a few minutes [1, 19, 20].

In particular, research on STLF is of special interest because it deals with the economic operation of power systems and other components of the energy mix, i.e. day-to-day operation of the power plants, the grid, and RESs, and electricity market pricing [1, 21]. Because of this importance, research in STLF is quite intense and many researchers are enthusiastic about the development of highly accurate models that can reliably predict energy demand for this time reference [16]. The study of STLF has been conducted using two main approaches, namely: statistical methods and machine learning-based methods. In the past, researchers relied heavily on statistical methods for load forecasting. Autoregressive models and time series methods have been applied to STLF in the literature concerning wind power generation, solar power generation, and electricity load forecasting [22]. However, these methods suffer from low accuracy, low sensitivity concerning input data as well and difficulty in incorporating complex realities such as weather data, intermittency of RES, and the dynamic behavior of EV for charging and V2G [22]. On the other hand, machine learning has been proven to be a viable approach to accurately carryout STLF and deal with complex situations and uncertainties.

Reviewed below are pertinent machine learning-based studies on STLF. In [23], the authors estimated the short hourly load consumption of Malaysia and the daily power electric consumption of Germany using a combination of long short-term memory networks (LSTM) and convolutional neural network (CNN) which was termed parallel LSTM-CNN Network. For the German data, the accuracy achieved was 91.18% while the accuracy for the Malaysia data achieved 98.23%. Moradzadeh et al. [24], utilized a form of deep learning approach using bidirectional long short-term memory (Bi-LSTM) to examine the short-term load forecasting of a microgrid on an hourly basis in rural sub-Saharan Africa. The training and testing results obtained from the Bi-LSTM network are 99.81% and 99.34%, respectively.

In a study by Solyali et al. [25], various machine learning techniques—including Artificial Neural Network (ANN), Neuro-Fuzzy Inference System (ANFIS), Multiple Linear Regression (MLR), and Support Vector Machine (SVM)—were evaluated for load prediction in Cyprus across short and long time frames. Utilizing variables like temperature, solar irradiation, humidity, population, GNI per capita, and electricity price per kWh, the research concluded that SVM excels in long-term energy generation forecasting, while ANN is more suitable for short-term analysis [26]. Notably, both SVM and ANN outperformed ANFIS and MLR in general load forecasting.

In [27], a study on short-term load forecasting in Qingdao, China utilized Bagged Regression Trees (BRT) and introduced indicator variables for special days like holidays. This approach improved BRT's performance and outperformed ANN in terms of speed and accuracy. In [28], Kernel Extreme Learning Machine (KELM) was applied to province-level STLF in China. By using kernelized principal component analysis, the authors reported KELM's superior performance over BPNN and ELM. Other authors [29] proposed three forecasting models—Multiple Linear Regression (MLR), Random Forest (RF), and Gradient Boosting (GB)—for day-ahead load demand in Southern California. GB was found to outperform both RF and MLR, with temperature and specific non-meteorological variables like holidays being the most influential factors.

In [30], a hybrid model was developed for the Queensland electric market, combining stationary wavelet packet transform and Harris Hawks optimization-based neural networks. The hybrid model showed better performance compared to traditional machine learning models such as PSO-based neural networks and least-square-support vector machines. In [31], a novel approach for capturing seasonality in day-ahead load forecasting was proposed. The proposed approach employs bidirectional Long Short-Term Memory (LSTM) Networks where an index governing the selection of seasonal models for forecasting purposes was derived from various weather patterns.

While numerous studies have explored short-term load forecasting, a gap exists in the literature concerning national-level forecasts that incorporate diverse energy sources. This research aims to fill this void by developing machine learning models for short-term electricity forecasting in Great Britain's power system, using a unique net imports dataset. This dataset, spanning 2010 to 2020, incorporates both renewable and conventional energy generation. The study is the first of its kind to apply machine learning techniques to a net imports dataset for Great Britain. We employed three machine learning algorithms—Artificial Neural Networks (ANN), Gaussian Process Regression (GPR), and Support Vector Regression (SVR) for both half-hourly and hourly forecasting. Model performance was assessed using Correlation Coefficient (R), Root Mean Square Error (RMSE), and Mean Absolute Deviation (MAD).

The primary novelty of this study is integrating a wide range of energy sources, including conventional, renewable, and interconnector inputs, into our short-term load forecasting models. This comprehensive approach is relatively unexplored in existing literature, particularly within the context of Great Britain's power system. Furthermore, the usual practice is to

partition the training to testing data in the ratio 70:30 or 80:20. This is usually done to allow the machine learning algorithm access to as many training examples as possible. In most instances, the machine learning algorithms can generalize over the training dataset and give excellent prediction accuracies when evaluated with the testing set [32, 33]. In other instances, the machine learning algorithms memorize the training set leading to over-fitting of the data and therefore leading to poor performance on the testing sets. In this investigation, we train the machine learning algorithms on a limited subset of data relative to the testing dataset. The training-testing split ratio for the half-hourly prediction models was 1:17 with 4000 data points for training and 69000 for testing. The split ratio for the half-hourly models was 1:8.5 with 4000 data points for training and 34000 for testing. This demonstrated the ability of the models to learn from relatively fewer data while being able to obtain good testing results with relatively larger data.

## Materials and methods

In this section, the data and features used for modeling the half-hourly and hourly load fore-casting problem are highlighted, as well as the description of the machine learning techniques used in this study.

### Data set

The modeling approach employed in this study uses the recently published data by [34], termed Elexon Sum Plus Embedded Net Imports (ESPENI). This dataset [35] is notably com-prehensive, spanning the period from 2009 to 2021, and embodies the amalgamation of two publicly accessible datasets, Elexon, and National Grid data, to formulate a unified dataset that approximates the aggregate electrical energy demand for Great Britain. Within this framework, the National Grid acts as the system operator, bearing the full responsibility for ensuring a balance between electricity generation and demand. Elexon, a subsidiary of the National Grid, primarily manages the flow of funds between the parties contributing to grid imbalance, either through demand or generation, and those that intervene to provide the requisite generation or load reduction to restore system balance. To accomplish these objec-tives, Elexon capitalizes on generation data sourced from various entities connected at the transmission levels. The ESPENI dataset exhibits several distinct characteristics, summarized as follows:

- The dataset illustrates the evolution in Great Britain's generation mix and electrical demand as depicted in Fig 1.

- It operates with a time granularity of 30 minutes, translating to 48 half-hourly settlement periods within a day.

- The dataset encompasses generation data from a diverse range of sources, inclusive of renewable energy sources.

- Data parsing is performed to furnish ISO-compatible UTC and local time values for each data entry.

- Missing data entries are substituted with values obtained through linear interpolation.

- The dataset is accessible under the CC-BY-NC license and is hosted on the Zenodo platform.

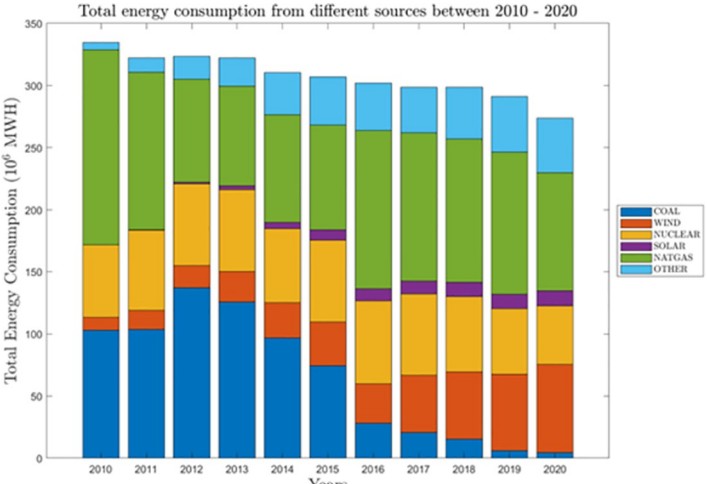

**Fig 1. Britain's energy consumption from different sources between 2010-2020.**

## Data preprocessing

The total estimated electricity consumption, sourced from the 'ESPENI_POWER_MW' column in the data set, was restructured to create regressors and target variables for both half-hourly and hourly forecasting. For forecasting on a half-hourly basis, the regressors encompass:

- Load observed in the preceding half-hour.

- Load observed during the identical half-hour on the preceding day.

- Load observed during the identical half-hour in the previous week.

- Load observed in the half-hour prior to the identical half-hour in the previous week.

- The day of the week.

- The time of the day.

    For forecasting on an hourly basis, the regressors include:

- Load observed in the previous hour.

- Load observed during the same hour on the preceding day.

- Load observed during the same hour in the previous week.

- Load observed in the hour preceding the same hour in the previous week.

- The day of the week.

- The time of the day.

## Support vector regression

Support Vector Regression (SVR) is a supervised learning algorithm mainly used for regression tasks. It operates by mapping input data into a high-dimensional feature space using a

kernel function. In this expanded space, SVR constructs an optimal hyperplane to make predictions [36]. The algorithm aims to minimize the error rate while optimizing a cost function to balance model complexity and prediction accuracy [37, 38]. Key parameters optimized include the regularization parameter C, the kernel type (linear, polynomial, RBF, etc.), and the kernel-specific parameters like $\gamma$ for the RBF kernel [39].

## Artificial neural networks

Artificial Neural Networks (ANNs) are inspired by biological neural systems and consist of interconnected nodes or "neurons" organized into layers. ANNs can model complex, non-linear relationships and are versatile enough for various learning tasks [28]. During training, the model adjusts the weights between nodes to minimize prediction errors [40, 41]. Parameters such as the learning rate, the number of hidden layers, and the number of neurons in each layer are commonly optimized [42, 43]. In this investigation, a feed-forward neural network model structure is utilized, paired with the Limited Memory Broyden-Fletcher-Goldfarb-Shanno quasi-Newton (LBFGS) training algorithm, aimed at minimizing the mean squared error criterion. The architectural layout of the neural network comprises five distinct layers: the input layer, a fully connected layer, a layer employing TRectifier Linear Unit (ReLU) activation, another fully connected layer, and finally, the output layer. The details of these are given in the results section.

## Gaussian Process Regression

Gaussian Process Regression (GPR) is a Bayesian, non-parametric approach employed for regression tasks, offering a probabilistic framework for both predictions and uncertainty quantification [44, 45]. It operates under the assumption that the dataset follows a multivariate Gaussian distribution. In this study, Bayesian optimization was used to fine-tune various hyperparameters and kernel functions for GPR. Key hyperparameters optimized include the basis function and sigma, along with the kernel function and its scale. The kernel functions considered for optimization include ARD exponential kernel and squared exponential kernel. For covariance between any two latent variables, $f(x_i)$ and $f(x_j) i \neq j, x, j \in R^d$ with kernel parameters $\theta$, the kernel functions $k(x_i, x_j | \theta)$ is defined as:

$$k(x_i, x_j | \theta) = \sigma_f^2 e^{-r}, \tag{1}$$

where $r = \sqrt{\sum_{m=1}^{d} \frac{(x_{im} - x_{jm})^2}{\sigma_m^2}}$ for the ARD exponential kernel and $r = \frac{1}{2} \frac{(x_i - x_j)^T (x_i - x_j)}{\sigma_l^2}$ for the squared exponential kernel, $\sigma_m$ is the length scale for predictor $m$: $m = 1, 2, 3, \ldots, d$, $\sigma_f$ is the standard deviation and $\sigma_l$ is the characteristic length scale.

## Bayesian optimization

Bayesian Optimization is an optimization algorithm designed for complex, costly, and non-linear functions [46–48]. It builds a probabilistic model of the objective function to predict both the expected value and uncertainty at new points. Parameters such as the acquisition function (e.g., Expected Improvement), exploration-exploitation trade-off, and kernel parameters for the Gaussian process model are optimized for efficient search [46–49].

## Model development

Figs 2 and 3 illustrate the workflow employed for developing the machine learning models for short-term load forecasting in this study. Initially, raw historical data was imported into

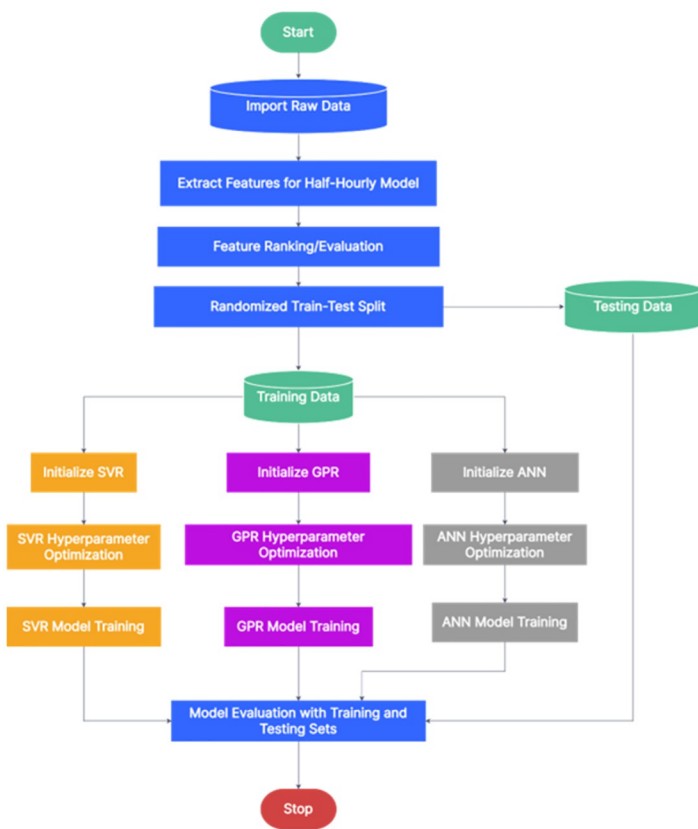

**Fig 2. Flowchart depicting the development of half-hourly forecasting models.**

MATLAB, where features relevant to both half-hourly and hourly load forecasting were extracted and evaluated. Subsequently, the data was randomized and partitioned into training and testing sets, with split ratios of 1:17 for half-hourly and 1:8.5 for hourly forecasting. Each machine learning algorithm was then initialized, and key hyperparameters were optimized using Bayesian optimization. Finally, the models were trained and evaluated using these segregated datasets.

## Results

This section deals with the results obtained from the load forecasting models. The outcomes are divided into two segments: half-hourly and hourly forecasting. The efficacy of the algorithms is assessed by employing metrics such as Root Mean Square Error (RMSE), Mean Absolute Deviation (MAD), Mean Absolute Percentage Error (MAPE) and Correlation of determination ($R^2$), as defined by Eqs (2) to (5), respectively.

$$RMSE = \sqrt{\frac{1}{n}\sum_{i=1}^{n}\left(y_i - \hat{y}_i\right)^2}, \tag{2}$$

$$MAD = \frac{1}{n}\sum_{i=1}^{n}|y_i - \hat{y}_i|, \tag{3}$$

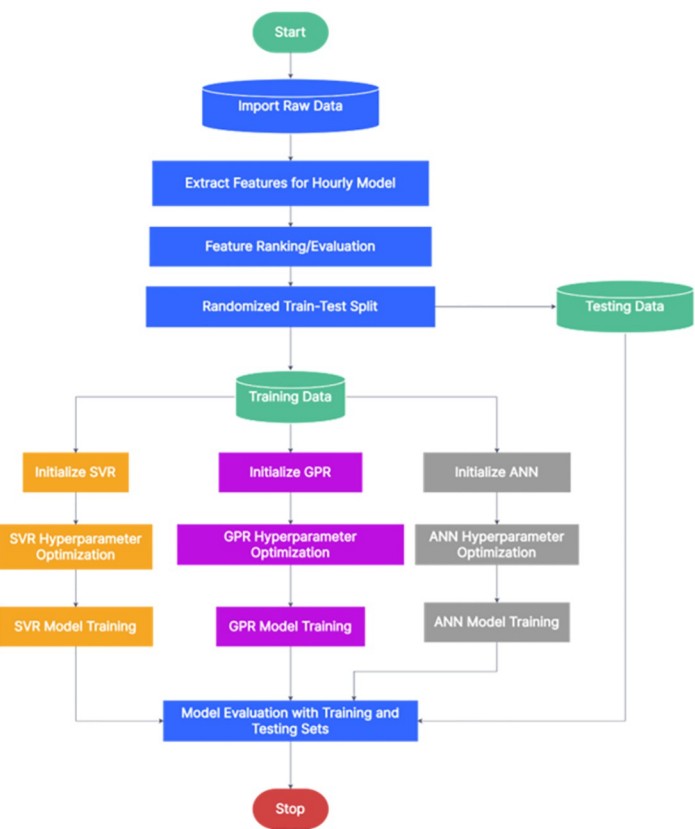

**Fig 3. Flowchart depicting the development of hourly forecasting models.**

$$MAPE = \frac{1}{n} \sum_{i=n}^{n} \left( \frac{|y_i - \hat{y}_i|}{y_i} \right) \quad (4)$$

$$R^2 = 1 - \frac{\sum (y_i - \hat{y}_i)^2}{\sum (y_i - \bar{y}_i)^2} \quad (5)$$

where $n$ represents the number of observations or forecasts, $y_i$ represents the actual values, $\bar{y}$ is the mean of the actual values and $\hat{y}_i$ represents the forecasted values.

## Half-hourly load forecasting

**Feature ranking.** Before training the machine learning models with the extracted features for predicting half-hourly demand, we were interested in evaluating each of the features to identify the importance of each of the selected features for the half-hourlmodelingng problem. For this task, we employed the univariate feature ranking function for regression using F-tests. All of the features identified for the half-hourly load forecasting problem showed high correlation, which implies that the best features were identified for this modelling problem.

**Hyperparameter optimization.** This section presents the results of the hyperparameter optimization of the machine learning algorithms for the half-hourly prediction models.

Table 1 shows the hyperparameters of the SVR, GPR, and ANN algorithms. The SVR algorithm converged to a linear kernel function with an epsilon of 8.4697 and a box constraint of 0.0010378. A pure quadratic basis function and an ard exponential kernel function were used for the GPR algorithm with a sigma value of 0.1848. In both instances, the standardized hyperparameter converged to a false value. The ANN algorithm converged to two hidden layers with 86 and 17 neurons respectively. The activation function in the hidden layers is the ReLU activation function while the lambda hyperparameter converged to $4.021 \times 10^{-6}$. In the case of the ANN algorithm, the standardized hyperparameter converged to a true value. The RMSE was used as the objective function. Fig 4 compares the minimum objective function evaluation over 30 iterations for each of the machine learning algorithms. It demonstrates that the GPR algorithm yielded the least minimum objective function value over the length of the optimization window.

**Modeling results.** In this section, we discuss the results derived from the models focused on half-hourly load forecasting. Table 2 provides a comprehensive summary of the performance metrics—RMSE, R, and MAD—for the SVR, GPR, and ANN algorithms across both

**Table 1. Hyperparameter optimization results.**

| SVR | | GPR | | ANN | |
|---|---|---|---|---|---|
| Parameter | Value | Parameter | Value | Parameter | Value |
| Box constraint | 0.0010378 | Sigma | 0.1848 | activations | ReLu |
| Kernel scale | - | kernel scale | - | Lambda | $4.021 \times 10^{-6}$ |
| Epsilon | 8.4697 | Basis function | pure quadratic | Layer weight initializer | Glorot |
| Kernel function | linear | Kernel function | ard exponential | Layer biases initializer | ones |
| Standardize | False | standardize | false | Standardize | true |
| Polynomial order | - | - | - | L1 | 86 |
| - | - | - | - | L2 | 17 |

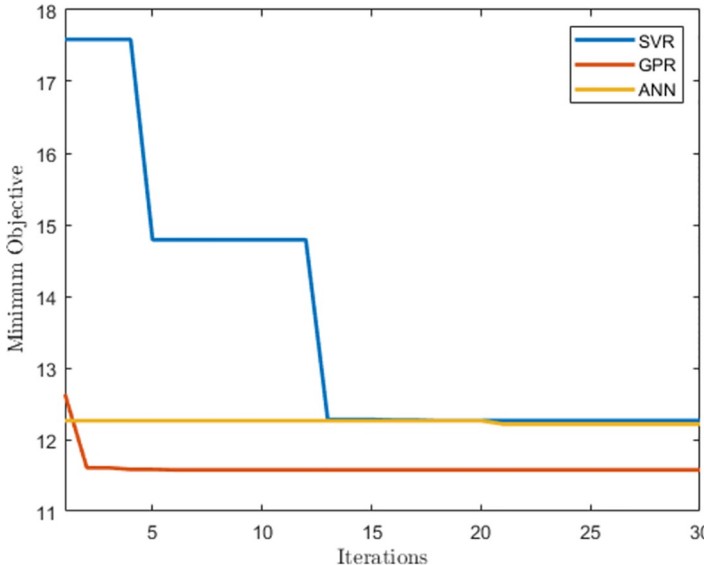

**Fig 4. Minimum objective function evaluations during hyperparameter optimization for half-hourly prediction models.**

**Table 2. Performance measures for half-hourly load forecasting.**

| | Training | | | | Testing | | | |
|---|---|---|---|---|---|---|---|---|
| | RMSE | $R^2$ | MAPE | MAD | RMSE | $R^2$ | MAPE | MAD |
| SVR | 461.73 | 0.995 | 0.0076 | 159.68 | 429.72 | 0.997 | 0.0096 | 159.708 |
| ANN | 417.30 | 0.996 | 0.0071 | 150.08 | 777.36 | 0.992 | 0.0188 | 291.39 |
| GP | 17.96 | 1 | 0.00039 | 6.714 | 487.02 | 0.996 | 0.0106 | 179.71 |

training and testing datasets. Figs 5–10 present the regression plots corresponding to SVR, ANN, and GPR models for both training and testing phases, respectively. In terms of training performance, GPR demonstrated superior results, achieving a 96.31% and 95.93% improvement in RMSE values over SVR and ANN models. Conversely, SVR excelled in the testing phase, registering a 44.72% and 11.77% reduction in RMSE values when compared to ANN and GPR models.

## Hourly load forecasting

**Feature ranking.** Six features were chosen for modeling the hourly demand: the load from the preceding hour, the load from the corresponding hour on the previous day, the load from the corresponding hour in the previous week, the load from the hour preceding the corresponding hour in the previous week, the day of the week, and the time of the day. The features again were assessed to evaluate their importance for the hourly demand forecasting problem using the univariate feature ranking function for regression based on F-tests. A high correlation was observed for all the features which once again confirms the ultimate suitability of the features for the forecasting problem.

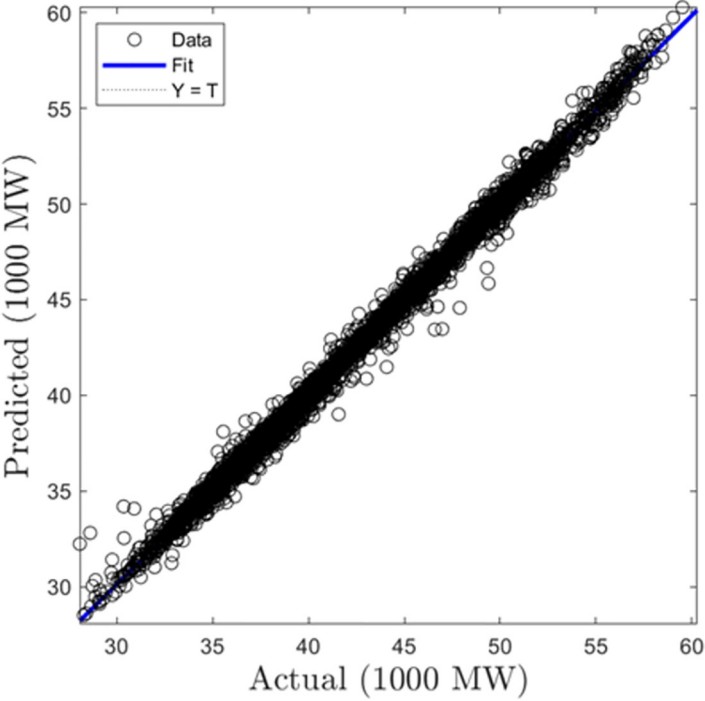

**Fig 5. SVR half-hourly training regression plots.**

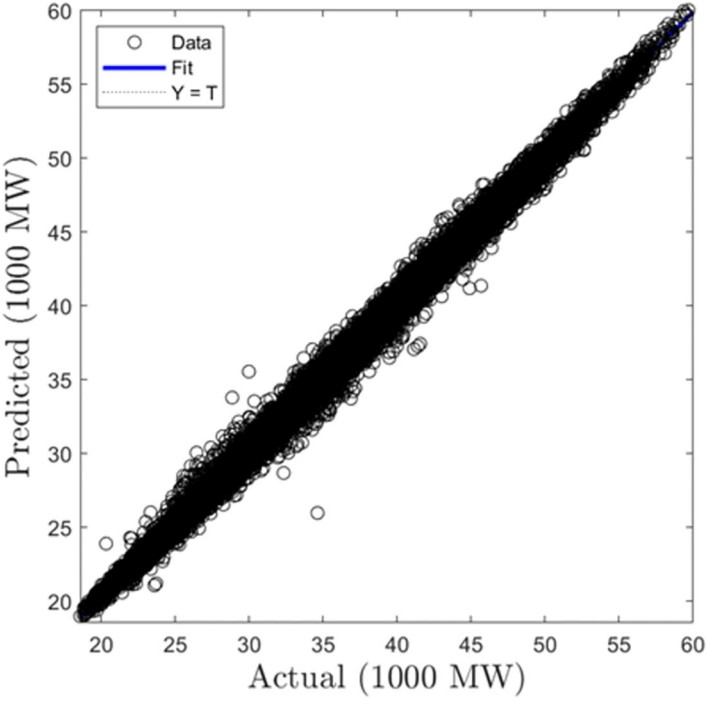

**Fig 6. SVR half-hourly testing regression plots.**

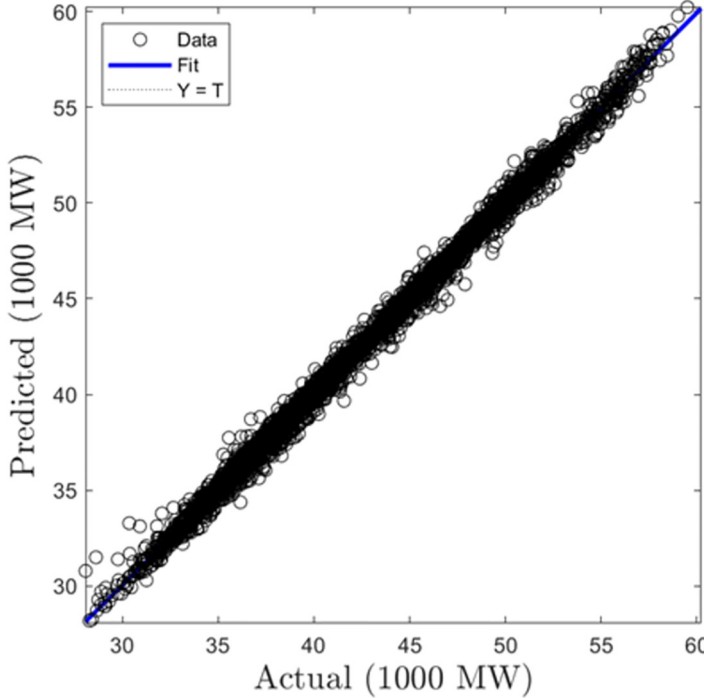

**Fig 7. ANN half-hourly training regression plots.**

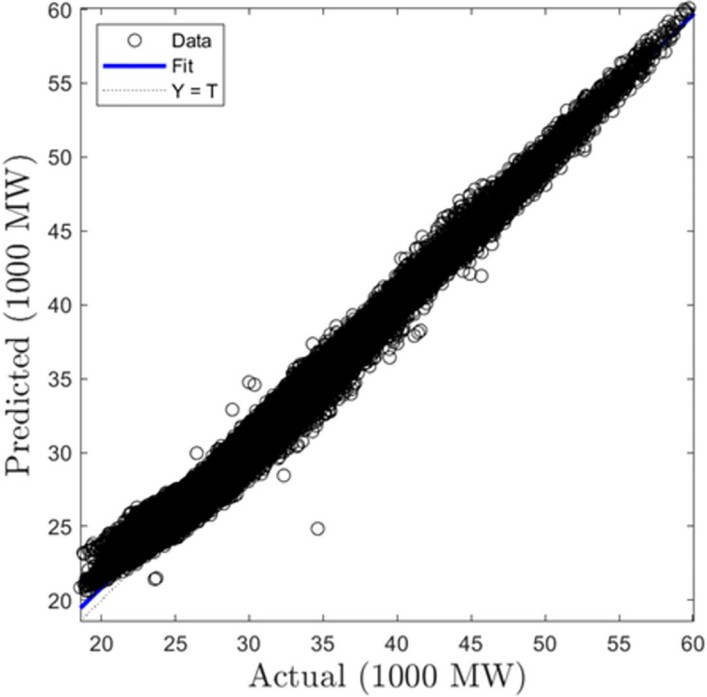

**Fig 8. ANN half-hourly testing regression plots.**

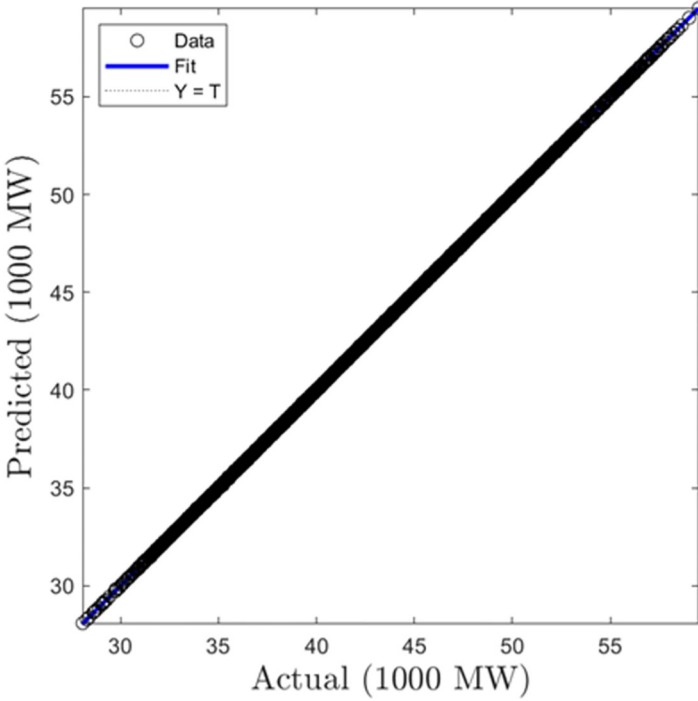

**Fig 9. GPR half-hourly training regression plots.**

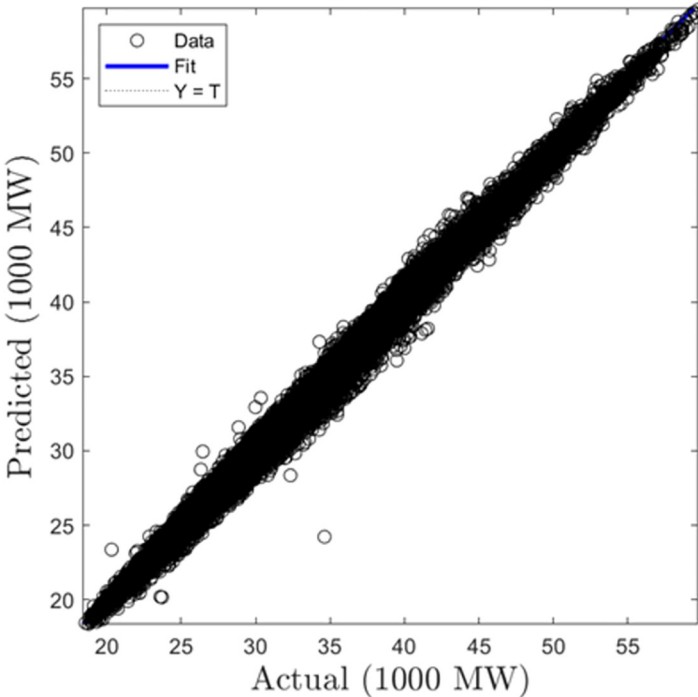

**Fig 10. GPR half-hourly testing regression plots.**

**Hyperparameter optimization.** This section presents the results of the hyperparameter optimization of the machine learning algorithms for the hourly prediction models. Table 3 shows the hyperparameters of the SVR, GPR and ANN algorithms respectively. The SVR algorithm converged to a polynomial kernel function of order 3. The epsilon and box constraint hyperparameters converged to 985.78 and 8.083, respectively. A linear basis function and an ard matern 3/2 kernel function with a sigma of 0.54071 were used for the GPR algorithm. The ANN algorithm converged to one hidden layer with 295 neurons respectively. The activation function in the hidden layer is the sigmoid activation function while the lambda hyperparameter converged to $4.021 \times 10^{-6}$. In the case of the ANN algorithm, the standardized hyperparameter converged to a true value. Fig 11 compares the minimum objective function evaluation over 30 iterations for each of the machine learning algorithms. Fig 11 demonstrates that the GPR algorithm yielded the least minimum objective function value over the length of the optimization window.

**Table 3. Hyperparameter optimization results.**

| SVR | | GPR | | ANN | |
|---|---|---|---|---|---|
| **Parameter** | **Value** | **Parameter** | **Value** | **Parameter** | **Value** |
| Box constraint | 985.78 | Sigma | 0.54071 | activations | sigmoid |
| Kernel scale | - | kernel scale | - | Lambda | $3.2357 \times 10^{-6}$ |
| Epsilon | 8.083 | Basis function | pure quadratic | Layer weight initializer | He |
| Kernel function | polynomial | Kernel function | ard matern 32 | Layer biases initializer | zeros |
| Standardize | true | standardize | false | Standardize | true |
| Polynomial order | 3 | - | - | L1 | 295 |

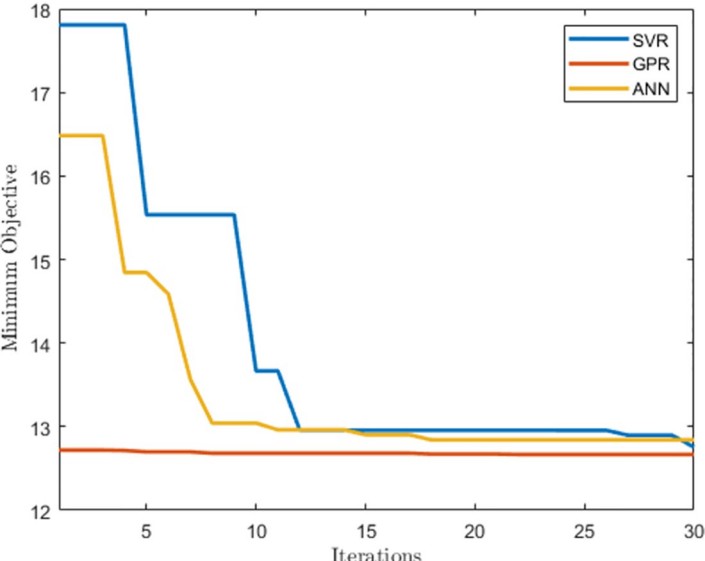

**Fig 11. Minimum objective function evaluations during hyperparameter optimization for hourly prediction models.**

**Modeling results.** In this section, we summarize the results obtained from the development of hourly forecasting models. Table 4 presents the values of the performance criterion recorded for all three machine learning algorithms for the half-hourly forecasting problem. Figs 12 and 13 display the regression plots for the training and testing phases of the hourly SVR load forecasting model, respectiely. Subsequently, Figs 14 to 17 show the regression plots for the training and testing phases of the ANN and GPR hourly forecasting models, respectively. On the training set, the GPR algorithm outperformed the SVR and ANN-based models with 96.62% and 95.99% improvement in RMSE values while the ANN algorithm showed a better generalization capability on the testing set with 4.09% and 9.78% better performance in RMSE values compared with SVR and GPR based models.

## Model comparisons

Finally, we compare the algorithms by generating forecasts for a typical day in 2010 and 2020. For this analysis, we select January 01, 2010, and January 01, 2020. In Figs 18 and 19, we show the forecast outputs obtained from the half-hourly forecasting models for both days compared to the actual load computed for those days. Likewise, Figs 20 and 21 shows the forecasts generated by all the algorithms for both days using the hourly forecasting model. Here we compared the algorithms using the mean absolute percentage error (MAPE) given by (4). The SVR, ANN, and GPR algorithms gave 1.67%, 1.42%, and 1.34% forecasting errors on the half-hourly

**Table 4. Performance measures for hourly load forecasting.**

| - | Training | | | | Testing | | | |
|---|---|---|---|---|---|---|---|---|
| - | RMSE | $R^2$ | MAPE | MAD | RMSE | $R^2$ | MAPE | MAD |
| SVR | 649.04 | 0.9922 | 0.0108 | 205.68 | 588.84 | 0.9936 | 0.0115 | 196.62 |
| ANN | 547.86 | 0.9938 | 0.0100 | 191.76 | 568.73 | 0.9942 | 0.0117 | 198.99 |
| GP | 21.97 | 1 | 0.0004 | 7.6305 | 625.94 | 0.9928 | 0.0134 | 625.94 |

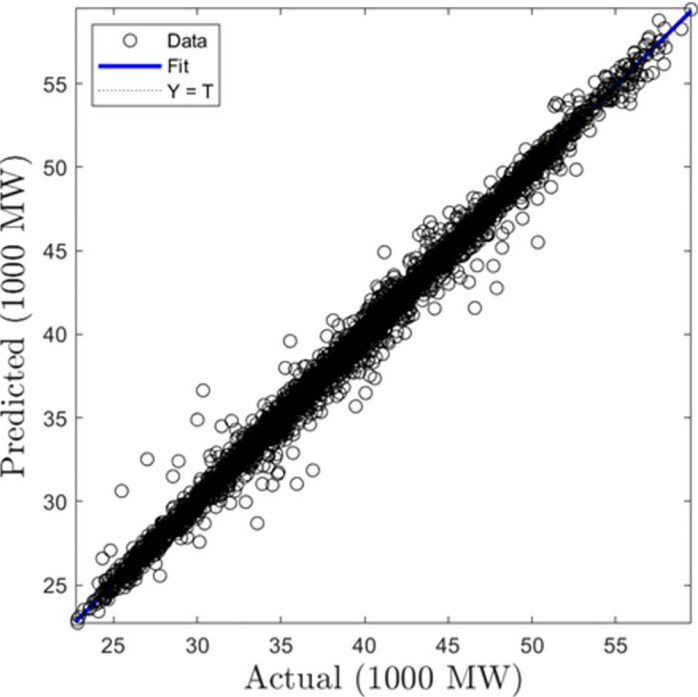

**Fig 12. SVR hourly trainingregression plots.**

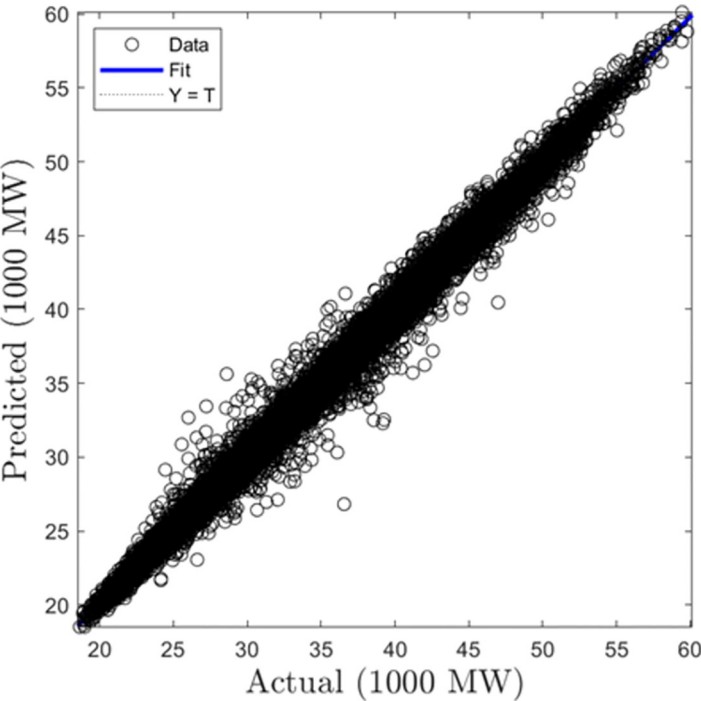

**Fig 13. SVR hourly testing regression plots.**

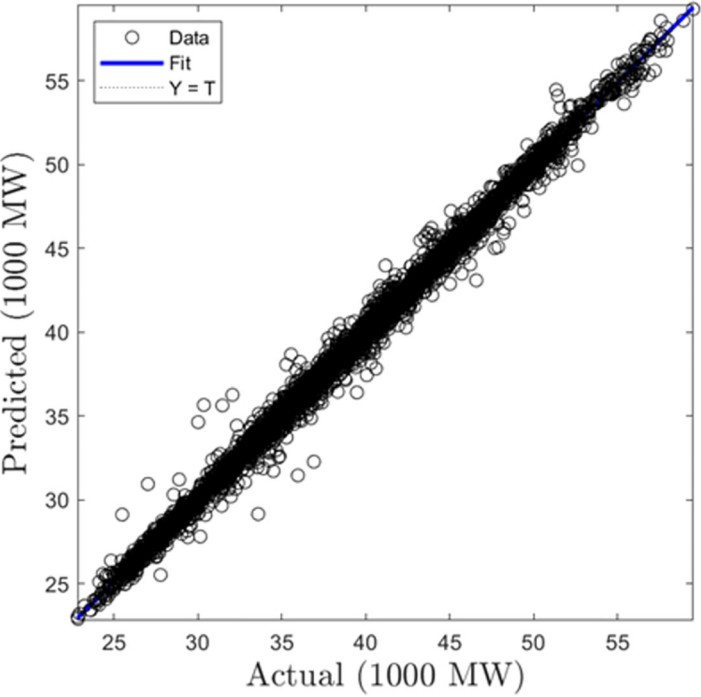

**Fig 14. ANN hourly training regression plots.**

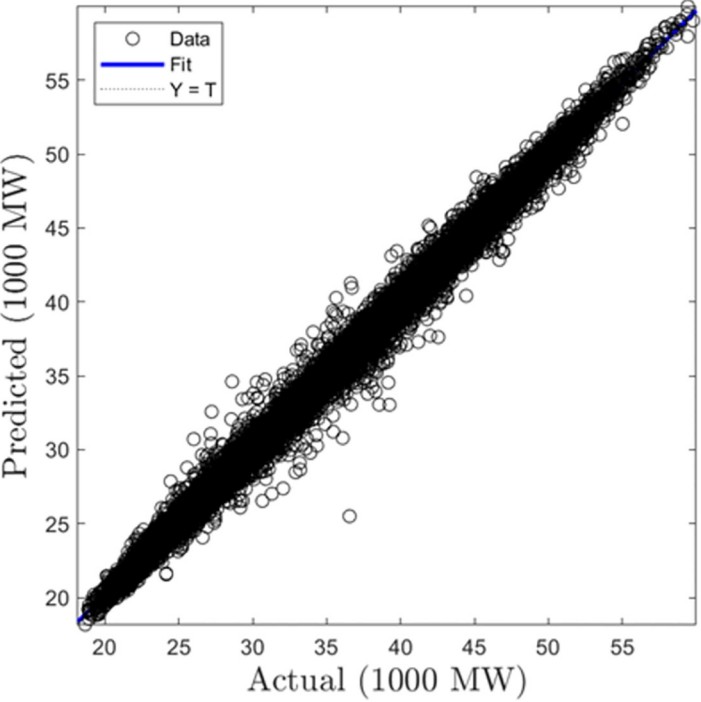

**Fig 15. ANN hourly testing regression plots.**

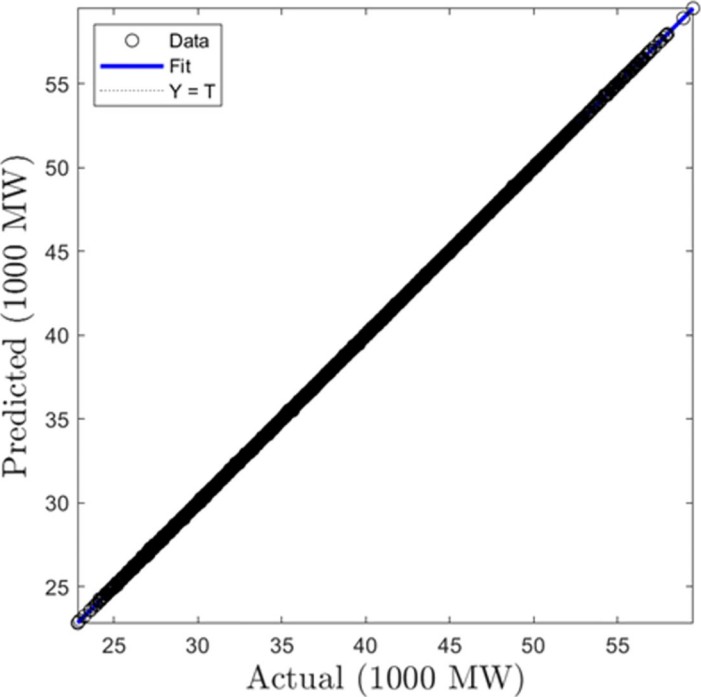

**Fig 16. GPR hourly training regression plots.**

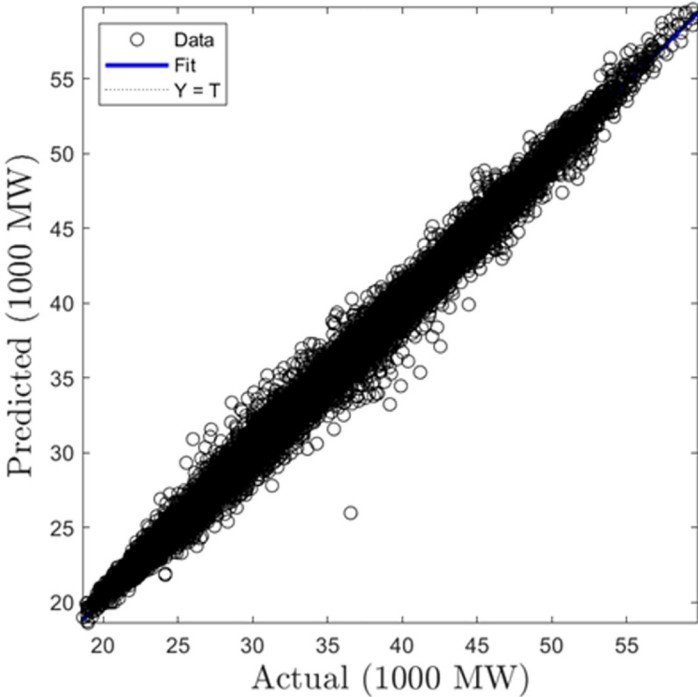

**Fig 17. GPR hourly testing regression plots.**

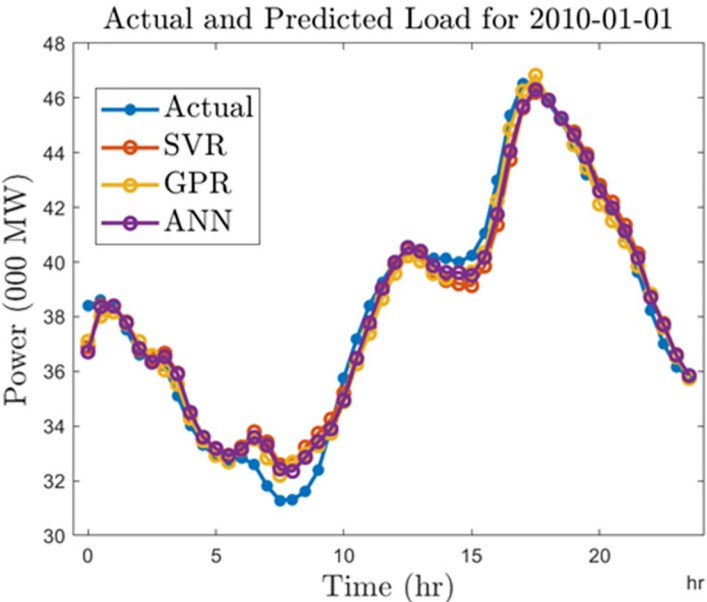

**Fig 18. Half-hourly predictions for a typical day in 2010.**

model on January 01, 2010, and 1.92%, 2.3%, and 1.49% on January 01, 2020. On the hourly forecasting models, 3.29%, 2.7%, and 3.05% forecasting errors were recorded for SVR, ANN, and GPR on January 01, 2010, while 3.32%, 2.99%, and 3.62% forecasting errors were recorded respectively for SVR, ANN, GPR algorithms on January 01, 2020.

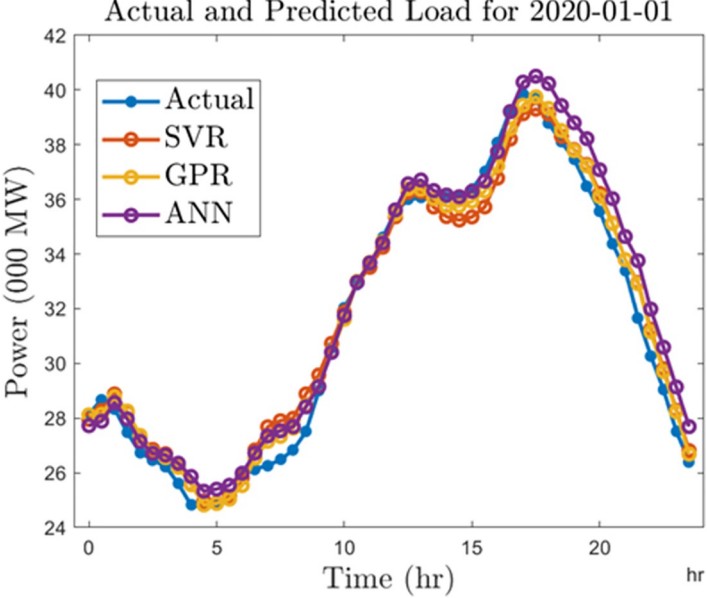

**Fig 19. Half-hourly predictions for a typical day in 2020).**

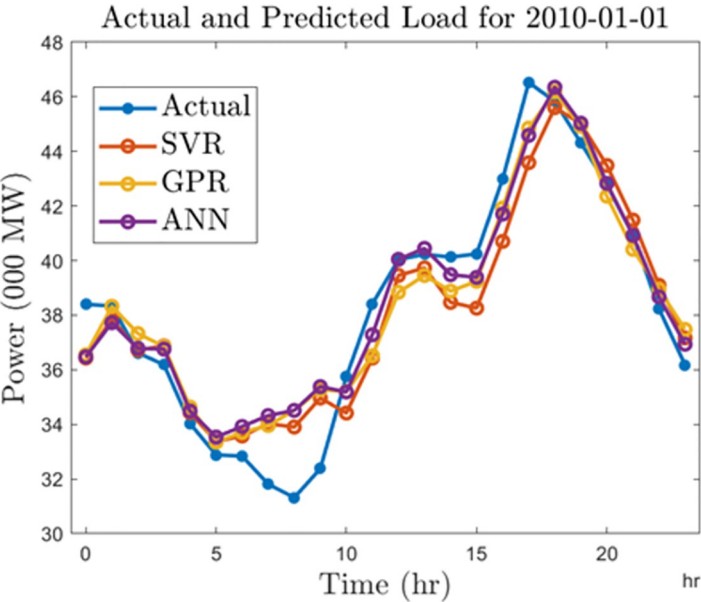

**Fig 20. Hourly predictions for a typical day in 2010.**

## Discussions

This study on short-term load forecasting in the UK's electrical grid, utilizing machine learning techniques and a comprehensive dataset, offers significant insights into energy management. The core achievement of this study lies in integrating various energy sources into the forecasting models, reflected in the enhanced accuracy and reliability of these models. The algorithms' efficacy, assessed through RMSE, MAD, and $R^2$, demonstrated notable

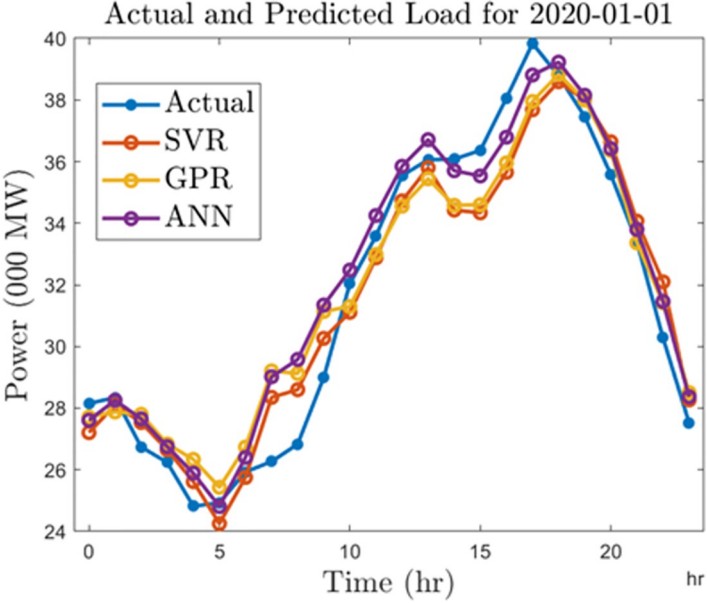

**Fig 21. Hourly predictions for a typical day in 2020.**

improvements in forecast precision. For instance, in half-hourly forecasting, GPR excelled in training, while SVR showed superior performance in the testing phases. Hourly forecasts saw SVR and ANN excel in different aspects of the modeling process. These improvements have direct implications for energy management and policy formulation. Accurate load forecasting aids in optimizing energy distribution and planning, which is crucial in transitioning to cleaner, renewable energy sources. The findings, particularly the algorithms' performance, underscore their real-world applicability and reliability in predicting energy needs. However, the study's focus on the UK's power system and the dataset from 2010 to 2020 may limit its applicability in other geographic settings with different energy mixes or consumption patterns. Furthermore, the complexity of machine learning models also poses challenges in terms of interpretability, which is vital for stakeholder communication and policy-making. Expanding the methodology to other regions and incorporating real-time data analytics will be a focal point of future research. This expansion will enhance the models' robustness and adaptability to different energy landscapes. Exploring advanced machine learning techniques like deep learning will provide a more robust understanding of energy demand forecasting in the geographical region of interest. These steps will contribute to creating more adaptable and comprehensive load forecasting models. Despite the specific focus on the UK's energy dataset, the methodologies and models used in this study have a high degree of adaptability. They can be tailored for various datasets and contexts, making these techniques widely applicable in different geographic and temporal settings. Data preprocessing, model selection, and validation principles demonstrated in this study are universally applicable across various machine-learning scenarios. This universality consolidates the potential global impact of the research and its contribution to advancing load forecasting techniques.

## Conclusions

The culmination of this study represents a significant advancement in short-term load forecasting, particularly within the dynamic context of the UK's evolving energy landscape. By integrating a diverse range of energy sources into the developed machine learning models, this research has responded to the critical need for more accurate and reliable forecasting methods and illuminated the path forward in energy management. The key insights from analyzing a decade's energy consumption data reveal a notable transition in the power generation mix. The decline in coal usage and the concurrent rise of wind and other renewable sources capture the essence of the ongoing transformation in the energy sector. While presenting unique challenges, this shift also opens up new opportunities for optimizing energy distribution and planning. The core of this study's contribution lies in the development and comparative analysis of the SVR, ANN, and GPR models. The remarkable accuracy of these models, as evidenced by their high correlation coefficients, underscores their potential as reliable tools for predicting short-term energy loads. More importantly, these models serve as a testament to the effectiveness of machine learning in adapting to and managing the complexities inherent in modern power systems. Furthermore, the methodologies and findings of this study have far-reaching implications, setting a precedent for similar research in other geographic regions. The adaptability of these models to different energy contexts demonstrates their relevance and applicability in a global setting, marking a substantial contribution to the field of energy management. In conclusion, this study not only bridges the gap identified at its inception but also paves the way for future innovations in load forecasting. It highlights the pivotal role of machine learning in navigating the complexities of today's energy systems and lays the groundwork for more sustainable, efficient, and responsive global energy management practices.

## Acknowledgments

The author acknowledges the support of the MIT Libraries.

## Author Contributions

**Conceptualization:** Yusuf A. Sha'aban.

**Data curation:** Yusuf A. Sha'aban.

**Formal analysis:** Yusuf A. Sha'aban.

**Funding acquisition:** Yusuf A. Sha'aban.

**Investigation:** Yusuf A. Sha'aban.

**Methodology:** Yusuf A. Sha'aban.

**Project administration:** Yusuf A. Sha'aban.

**Resources:** Yusuf A. Sha'aban.

**Software:** Yusuf A. Sha'aban.

**Validation:** Yusuf A. Sha'aban.

**Visualization:** Yusuf A. Sha'aban.

**Writing – original draft:** Yusuf A. Sha'aban.

**Writing – review & editing:** Yusuf A. Sha'aban.

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
