## [Decision Letter · Decision Letter 0]

8 Dec 2023

PONE-D-23-39140Predictive models for short-term load forecasting in the UK’s electrical gridPLOS ONE

Dear Dr. Sha'aban,

Thank you for submitting your manuscript to PLOS ONE. After careful consideration, we feel that it has merit but does not fully meet PLOS ONE’s publication criteria as it currently stands. Therefore, we invite you to submit a revised version of the manuscript that addresses the points raised during the review process.

We look forward to receiving your revised manuscript.

Kind regards,

Sathishkumar Veerappampalayam Easwaramoorthy

Academic Editor

PLOS ONE

Journal Requirements:

4. Please ensure that you refer to Figure 4 in your text as, if accepted, production will need this reference to link the reader to the figure.

Reviewers' comments:

Reviewer's Responses to Questions

**Comments to the Author**

1. Is the manuscript technically sound, and do the data support the conclusions?

Reviewer #1: Yes

Reviewer #2: Partly

Reviewer #3: Yes

2. Has the statistical analysis been performed appropriately and rigorously? 

Reviewer #1: Yes

Reviewer #2: Yes

Reviewer #3: Yes

3. Have the authors made all data underlying the findings in their manuscript fully available?

Reviewer #1: Yes

Reviewer #2: No

Reviewer #3: Yes

4. Is the manuscript presented in an intelligible fashion and written in standard English?

Reviewer #1: Yes

Reviewer #2: Yes

Reviewer #3: Yes

5. Review Comments to the Author

Reviewer #1: what is (000MW) in all related figures?

quality of figures are poor.

with the regression process you have only shown R or Rsquare. What is the governing equation that lead to least errors in this case.

why your algorithm is performing better than the other algos?

clearly define all parameters listed in all equations.

what is novelty that you want to claim?

what is objective function in this case?

Reviewer #2: TITLE: Predictive models for short-term load forecasting in the UK’s electrical grid

Manuscript Number : PONE-D-23-39140_reviewer

Type of manuscript : Original Article

Summary

In this paper, the authors used machine learning models for short-term load forecasting in Great Britain's power system using a unique data set incorporating generation, conventional, renewable, and interconnectors to approximate total electricity demand.

I found this paper interesting and important in the energy sector. However, the manuscript requires some improvements. Some general comments are:

General Comments

• In the Abstract, the authors should briefly present and justify the novelty of their approach in contrast to the existing body of knowledge.

• Provide a list of abbreviations used in the paper.

• What is the justification of the evaluation metrics used in this study, RMSE, MAD, MAPE and R? It is also usually recommended to use MASE when modelling time series data. How did you select the best kernel for each SVR and GPR?

• Include a separate “Discussion” section. In this section, the authors should present their findings and their main implications, highlight their study's limitations, and briefly mention some precise directions they intend to follow in their future research work. Can the authors mention how much of their research is influenced by the data they used or to which extent the methodology used within the developed research can be easily applied to other situations when the datasets differ? In this way, the authors could highlight their approach's generalisation capability to justify a wider contribution to the current state of the art.

• In the “Conclusion” section, authors should avoid summarising the aspects they have already stated in the body of the manuscript. Instead, they should interpret their findings at a higher level of abstraction than in the previous sections of the manuscript. The authors should highlight whether or to what extent they have addressed the necessity identified within the "Introduction" section (the identified gap). The authors should avoid restating everything they did once again. However, instead, they should emphasise what their findings mean to the readers, making the "Conclusions" section interesting and memorable to them. The authors should not restate what they have done or what the article does. They should focus instead on what they have discovered and, most importantly, on what their findings mean.

Reviewer #3: ---Write the limitations of the methods.

---Add the limitations of the study and your suggestions to the results.

---Test the reliability of prediction results. See the article below for reliability:

Sun, J., Zhang, J., Wang, J., Liang, J., Zhang, S., Chen, S., Le, J., Rui, X., Ge, S., Li, L., 2008. Standard for hydrological information and hydrological forecasting. Chinese Stand.

---Determine whether the prediction results have the same mean as the actual data. Perform the Kruskal-Wallis test.

---Why is there no discussion section?

---Use R2 instead of R.

---Add the MAPE criterion to the table 4.

---There is no need for scatter plots of the training data.

---Draw a Taylor diagram to compare the methods.

---Check out the following articles about ANN, GPR and SVM:

Başakın, E. E., Ekmekcioğlu, Ö., Çıtakoğlu, H., & Özger, M. (2022). A new insight to the wind speed forecasting: robust multi-stage ensemble soft computing approach based on pre-processing uncertainty assessment. Neural Computing and Applications, 34(1), 783-812.

Uncuoglu, E., Citakoglu, H., Latifoglu, L., Bayram, S., Laman, M., Ilkentapar, M., & Oner, A. A. (2022). Comparison of neural network, Gaussian regression, support vector machine, long short-term memory, multi-gene genetic programming, and M5 Trees methods for solving civil engineering problems. Applied Soft Computing, 129, 109623.

Citakoglu, H. (2021). Comparison of multiple learning artificial intelligence models for estimation of long-term monthly temperatures in Turkey. Arabian Journal of Geosciences, 14, 1-16.

Citakoglu, H., & Coşkun, Ö. (2022). Comparison of hybrid machine learning methods for the prediction of short-term meteorological droughts of Sakarya Meteorological Station in Turkey. Environmental Science and Pollution Research, 29(50), 75487-75511.

Demir, V., & Citakoglu, H. (2023). Forecasting of solar radiation using different machine learning approaches. Neural Computing and Applications, 35(1), 887-906.

6. PLOS authors have the option to publish the peer review history of their article (what does this mean?). If published, this will include your full peer review and any attached files.

Reviewer #1: **Yes: **khurram saleem alimger

Reviewer #2: No

Reviewer #3: No

---

## [Author Response · Author response to Decision Letter 0]

23 Dec 2023

The response to the reviwers have been uploaded as a PDF docuemnt

---

## [Decision Letter · Decision Letter 1]

3 Jan 2024

Predictive models for short-term load forecasting in the UK’s electrical grid

PONE-D-23-39140R1

Dear Dr. Sha'aban,

We’re pleased to inform you that your manuscript has been judged scientifically suitable for publication and will be formally accepted for publication once it meets all outstanding technical requirements.

Kind regards,

Sathishkumar Veerappampalayam Easwaramoorthy

Academic Editor

PLOS ONE

Additional Editor Comments (optional):

Reviewers' comments:

Reviewer's Responses to Questions

**Comments to the Author**

1. If the authors have adequately addressed your comments raised in a previous round of review and you feel that this manuscript is now acceptable for publication, you may indicate that here to bypass the “Comments to the Author” section, enter your conflict of interest statement in the “Confidential to Editor” section, and submit your "Accept" recommendation.

Reviewer #1: All comments have been addressed

Reviewer #2: All comments have been addressed

Reviewer #3: All comments have been addressed

2. Is the manuscript technically sound, and do the data support the conclusions?

Reviewer #1: Yes

Reviewer #2: Yes

Reviewer #3: Yes

3. Has the statistical analysis been performed appropriately and rigorously? 

Reviewer #1: Yes

Reviewer #2: Yes

Reviewer #3: Yes

4. Have the authors made all data underlying the findings in their manuscript fully available?

Reviewer #1: Yes

Reviewer #2: No

Reviewer #3: Yes

5. Is the manuscript presented in an intelligible fashion and written in standard English?

Reviewer #1: Yes

Reviewer #2: Yes

Reviewer #3: Yes

6. Review Comments to the Author

Reviewer #1: figure quality is not improved. However, if the editor agrees then it is fine with me.

Reviewer #2: The authors have fully addressed all my comments in the previous review report I submitted. I recommend that the paper be accepted for publication.

Reviewer #3: (No Response)

7. PLOS authors have the option to publish the peer review history of their article (what does this mean?). If published, this will include your full peer review and any attached files.

Reviewer #1: **Yes: **Khurram Saleem Alimgeer

Reviewer #2: No

Reviewer #3: No

---

## [Editor Report · Acceptance letter]

11 Jan 2024

PONE-D-23-39140R1 

PLOS ONE

Dear Dr. Sha'aban, 

I'm pleased to inform you that your manuscript has been deemed suitable for publication in PLOS ONE. Congratulations! Your manuscript is now being handed over to our production team.

Kind regards, 

on behalf of

Dr. Sathishkumar Veerappampalayam Easwaramoorthy 

Academic Editor

PLOS ONE